# Bayesian approach to assessing population differences in genetic risk of disease with application to prostate cancer

**Iain R. Timmins**[1,2,3]*, **The PRACTICAL Consortium**[¶], **Frank Dudbridge**[1]

**1** Department of Population Health Sciences, University of Leicester, Leicester, United Kingdom, **2** Division of Genetics and Epidemiology, The Institute of Cancer Research, London, United Kingdom, **3** Statistical Innovation, AstraZeneca, Cambridge, United Kingdom

¶ Membership of the PRACTICAL consortium is provided in Supporting Information file
S1 Acknowledgements.
* iain.timmins@btinternet.com

**Data Availability Statement:** The summary statistics on prostate cancer are currently available via dbGaP at phs001120.v2.p2, or via application to PRACTICAL (contact: PRACTICAL@icr.ac.uk).

## Abstract

Population differences in risk of disease are common, but the potential genetic basis for these differences is not well understood. A standard approach is to compare genetic risk across populations by testing for mean differences in polygenic scores, but existing studies that use this approach do not account for statistical noise in effect estimates (i.e., the GWAS betas) that arise due to the finite sample size of GWAS training data. Here, we show using Bayesian polygenic score methods that the level of uncertainty in estimates of genetic risk differences across populations is highly dependent on the GWAS training sample size, the polygenicity (number of causal variants), and genetic distance $(F_{ST})$ between the populations considered. We derive a Wald test for formally assessing the difference in genetic risk across populations, which we show to have calibrated type 1 error rates under a simplified assumption that all SNPs are independent, which we achieve in practise using linkage disequilibrium (LD) pruning. We further provide closed-form expressions for assessing the uncertainty in estimates of relative genetic risk across populations under the special case of an infinitesimal genetic architecture. We suggest that for many complex traits and diseases, particularly those with more polygenic architectures, current GWAS sample sizes are insufficient to detect moderate differences in genetic risk across populations, though more substantial differences in relative genetic risk (relative risk > 1.5) can be detected. We show that conventional approaches that do not account for sampling error from the training sample, such as using a simple $t$-test, have very high type 1 error rates. When applying our approach to prostate cancer, we demonstrate a higher genetic risk in African Ancestry men, with lower risk in men of European followed by East Asian ancestry.

## Author summary

Many diseases and complex traits, such as prostate cancer, exhibit differences in incidence across populations. Yet the potential contribution of genetic factors towards such

**Funding:** FD is funded by the MRC (MR/S037055/1, https://mrc.ukri.org/). The funders had no role in study design, data collection and analysis, decision to publish, or preparation of the manuscript.

**Competing interests:** IRT is a full-time employee of AstraZeneca.

disparities is unclear. Polygenic scores summarise genetic effects across the genome and can in principle provide a valuable tool for assessing and comparing disease risk across populations. In practise, current approaches based on polygenic scores assume that such scores perfectly measure genetic risk of disease without measurement error, and thus do not account for uncertainty that arises in the construction of the score from a finite genome-wide association study (GWAS) training sample, which can be substantial. We introduce a Bayesian approach based on the LDpred2 polygenic score model that accounts fully for training sample uncertainty, and we propose a Wald test for formally testing such genetic risk differences across populations. Simulations show that the method properly controls for type 1 errors assuming independent SNPs (achieved by pruning), and that statistical power is sensitive to both the genetic architecture (heritability and polygenicity) and training sample size. In application to prostate cancer, this framework enables us to identify a higher genetic risk in African Ancestry men, with lower risk in men of European followed by East Asian ancestry.

## Introduction

An important open question in genetics is the degree to which population differences in disease risk can be attributed to differences in genetic risk. This question has recently been considered for several diseases and complex traits [1–7] including prostate cancer [8].

One commonly performed approach uses $t$-tests and ANOVA to examine differences in mean polygenic scores. Using this method, Conti et al. [8] estimated a 2.18-times higher genetic risk for prostate cancer in African men in comparison with European ancestry men, and a 0.73-times lower genetic risk in East Asian compared to European ancestry men. Similarly, Morris et al. [9] found evidence of a higher genetic risk of lupus in East Asian compared to most other global populations. In each case, using polygenic scores, the authors have drawn strong conclusions about differences in disease risk being attributable to different underlying genetic risk profiles.

However, we argue that these approaches are problematic, because they do not account for uncertainty that arises in the construction of the polygenic score itself. In effect, using $t$-tests and ANOVA approaches in this manner assumes that polygenic scores measure genetic risk of disease without any measurement error. This would be appropriate for estimating the accuracy of a given polygenic score between populations, such as to assess prospects for risk prediction. However for inference about the underlying genetic risk, we must allow that polygenic scores are derived from a finite sample of GWAS training data, and as shown previously [10–12], genetic risk estimates based on polygenic scores carry large amounts of uncertainty which should be accounted for in subsequent analyses.

To better understand and explain this source of uncertainty from the GWAS training data, consider a scenario where we repeatedly perform a GWAS analysis, and then create polygenic scores and calculate their mean difference between two population samples. Then across such repeats there will be variation in the estimates of genetic risk difference. This variation will, among other factors, be dependent upon the initial GWAS sample size. Since previous studies do not account for such inferential variation from the GWAS training data, one can question the certainty of conclusions drawn.

To address this issue, we develop a framework for estimating the genetic contribution to the relative risk of disease across populations that accounts for uncertainty in the construction of the polygenic score. We examine how the uncertainty in estimates of relative genetic risk

across populations depends on the genetic architecture of the disease, the training sample size, and the genetic distance ($F_{ST}$) between populations compared. We further derive a Wald test statistic for formally testing differences in genetic risk across populations, which we show to have good control of type 1 errors under a simplified assumption of independent SNPs, which in practise we achieve through a pruning approach. We stress that future research should seek to extend this method further to account for the realistic effects of linkage disequilibrium (LD) and imperfect tagging. Finally, we also re-analyse GWAS summary data from Conti et al. [8] on prostate cancer using a genome-wide polygenic score derived using LDpred2 [13], ensuring that uncertainty in the training sample is taken into account.

## Description of the Method

### Bayesian approach to estimating relative risk for populations

We use a Bayesian approach to estimating relative genetic risk across populations. Let $y_i$ be a trait measured on the $i$th individual, $x_i$ an $M \times 1$ vector of genotypes (equal to 0, 1 or 2) and $\boldsymbol{\beta}$ an $M \times 1$ vector of per-allele effects for each genetic variant. We consider a linear model $y_i = x_i^T \boldsymbol{\beta} + \epsilon_i$. The genetic effects are estimated as $\hat{\boldsymbol{\beta}}_{\text{GWAS}}$, denoting the marginal estimates from GWAS summary statistics. We use the Bayesian approach of LDpred2 [13], which assumes the effects at SNP $j$ are drawn from a mixture distribution, where $f_j$ denotes the allele frequency in the training sample:

$$\beta_j \sim \begin{cases} N\left(0, \dfrac{h_g^2}{M p_{\text{causal}}[2f_j(1-f_j)]}\right), & \text{with probability } p_{\text{causal}} \\ 0, \text{with probability } 1 - p_{\text{causal}} \end{cases}$$

By combining the prior distribution $p(\boldsymbol{\beta}|h_g^2, p_{\text{causal}})$ and the likelihood of the observed data $p(\hat{\boldsymbol{\beta}}_{\text{GWAS}}|\boldsymbol{\beta})$, we can compute a posterior distribution as $p(\boldsymbol{\beta}|\hat{\boldsymbol{\beta}}_{\text{GWAS}}; h_g^2, p_{\text{causal}})$. We use $\tilde{\boldsymbol{\beta}} \sim p(\boldsymbol{\beta}|\hat{\boldsymbol{\beta}}_{\text{GWAS}}; h_g^2, p_{\text{causal}})$ to refer to the samples from the posterior distribution. The genetic value for individual $i$ is estimated through the Bayesian polygenic score, defined as $\hat{PGS}_i = x_i^T \mathbb{E}[\boldsymbol{\beta}|\hat{\boldsymbol{\beta}}_{\text{GWAS}}; h_g^2, p_{\text{causal}}]$.

Suppose further that we wish to find the relative genetic risk between two populations. We assume that the genetic effects are the same in both populations, so that differences in genetic risk arise only from differences in allele frequencies. Let the allele frequencies for variant $j$ be $f_j$ and $g_j$ in populations 1 and 2. Using the result derived by Conti et al. [8], the relative risk comparing population 1 versus population 2 is $RR = \exp[d]$, where the difference $d$ in mean genetic risk (on the log scale) across the populations is given by:

$$d = \mathbb{E}[x_1^T \boldsymbol{\beta}] - \mathbb{E}[x_2^T \boldsymbol{\beta}] = \sum_{j=1}^M 2(f_j - g_j)\beta_j$$

Next, we demonstrate that the posterior variance, $\text{var}[d|\hat{\boldsymbol{\beta}}_{\text{GWAS}}; h_g^2, p_{\text{causal}}]$ can be evaluated analytically under a simplified assumption where all SNPs are assumed independent. Under the non-infinitesimal model, we have derived analytical expressions for the posterior variance of individual SNP effect sizes $\text{var}[\beta_j|\hat{\beta}_{\text{GWAS},j}, h_g^2, p_{\text{causal}}]$ (see S1 Appendix). Using these, and

under the assumption of independent SNPs, the posterior variance of $d$ can be expressed as:

$$\text{var}[d|\hat{\boldsymbol{\beta}}_{\text{GWAS}}; h_g^2, p_{\text{causal}}] = \sum_1^M 4(f_j - g_j)^2 \text{var}[\beta_j|\hat{\beta}_{\text{GWAS},j}; h_g^2, p_{\text{causal}}] \tag{1}$$

We demonstrate the validity of our analytical expressions for the posterior variance of $d$ through comparisons with LDpred2-auto for a range of genetic architectures and GWAS training sample sizes (S1 Fig).

For the case of an infinitesimal model, we show further that the analytical form can be approximated by a simple closed-form expression (see S2 Appendix):

$$\text{var}[d|\hat{\boldsymbol{\beta}}_{\text{GWAS}}, h_g^2] \approx \frac{4MF_{ST}}{N} \left(1 + \frac{M}{Nh_g^2}\right)^{-1} \tag{2}$$

We demonstrated the validity of the closed-form analytical expressions for the posterior variance of $d$ through comparisons with LDpred2-grid (with $p_{\text{causal}}$ set to 100%), showing the formula has the correct functional dependence with each of the parameters $h_g^2$, $M$, $N$ and $F_{ST}$ (S2 Fig).

Additionally, to test for differences in genetic risk across populations, we consider a Wald test on the posterior mean estimator $\hat{d} = \mathbb{E}[d|\hat{\boldsymbol{\beta}}_{\text{GWAS}}; h_g^2, p_{\text{causal}}]$ (S3 Appendix).

## Verification and comparison

### Simulation study

We simulated a range of genetic architectures and training sample sizes to understand the expected levels of uncertainty in estimates of $d$. To ensure our simulations were tractable at large samples, we simulated GWAS summary statistics, rather than individual-level phenotypes and genotypes.

Summary statistics were simulated using a wide range of values of SNP-heritability $h_g^2$, polygenicity $p_{\text{causal}}$, genetic differentiation $F_{ST}$, and effective training sample size $N_{\text{eff}}$. We assumed the genetic architecture could be described by a set of $M = 200{,}000$ SNPs, which were independent (i.e., in linkage equilibrium) and directly genotyped in both populations. This best-case scenario gives a lower bound on the uncertainty expected in real data applications.

We generated allele frequencies using a version of the Balding-Nichols model [14] where we assume that the ancestral allele frequency is unknown [15]. We assume a simplified model where all SNPs are independent. For each SNP, the allele frequency $f_j$ in the first population was drawn from the uniform distribution on [0.1,0.9], and the allele frequency $g_j$ of the second populations was drawn from a beta distribution with parameters $f_j(1-2F_{ST})/2F_{ST}$ and $(1-f_j)(1-2F_{ST})/2F_{ST}$.

We considered a range of genetic architectures: a heritability $h_g^2$ of 0.05, 0.10, 0.25, 0.50, 0.80, polygenicity $p_{\text{causal}}$ of 0.1%, 1%, 10%, and 100%, genetic differentiation $F_{ST}$ of 0.02, 0.04, 0.06, 0.08, 0.10 and 0.12, and we varied the effective sample size $N_{\text{eff}}$ between $10^3$ and $10^8$.

For each simulation replicate, SNP effect sizes were drawn from the non-infinitesimal mixture model, based on the allele frequencies of population 1:

$$\beta_j \sim \begin{cases} N\left(0, \dfrac{h_g^2}{Mp_{\text{causal}}[2f_j(1-f_j)]}\right), & \text{with probability } p_{\text{causal}} \\ 0, \text{with probability } 1 - p_{\text{causal}} \end{cases}$$

and marginal effect estimates were drawn from [16]:

$$\hat{\beta}_{\mathrm{GWAS},j}|\beta_j \sim N\left(\beta_j, \frac{1}{2Nf_j(1-f_j)}\right)$$

where it was assumed that the effect sizes are estimated based on samples predominantly taken from population 1. We used the derived analytical expressions to efficiently evaluate the posterior mean and variance of $d$ without recourse to MCMC simulations, where we set $p_{\mathrm{causal}}$ and $h_g^2$ equal to their true values. For each set of parameters, we estimated the expected posterior variance of $d$ by averaging results across 100 summary statistic replicates. The steps for performing this simulation study are provided in S5 Appendix.

Additionally, for each set of parameters, we evaluated the expected sampling variance, bias and mean square error of the posterior mean $\hat{d}$, also by averaging across 100 simulations. The steps of this simulation study evaluating the properties of $\hat{d}$ are provided in S6 Appendix.

To understand the type 1 error for our Wald test on $\hat{d}$, we generated $\chi^2$-test statistics for each simulation under the null hypothesis $d = 0$. We also evaluated the statistical power of the Wald test, where we simulated from scenarios corresponding to a relative risk ($RR = \exp[d]$) of 1.1, 1.2, 1.5 and 2, whilst varying both the training sample size and genetic architecture. We also evaluated the type 1 error of the $t$-test, where we varied the target sample size between 5,000, 10,000, 20,000, and 100,000 for both of the two target populations. For statistical power and type 1 error calculations, we used 1,000 simulation replicates. The steps of this simulation study for evaluating the type 1 error and power of the Wald test are also provided in S6 Appendix.

## Results

The main expressions derived above are the Bayesian posterior variance of the difference in polygenic scores between populations, and in particular, the closed-form expression for this variance under an infinitesimal model. Here we explore the properties of the posterior variance and use simulations to confirm the derived expressions.

### Posterior variance of genetic relative risk

We illustrate the impact of the training sample size and genetic architecture on estimating the genetic relative risk ($RR = \exp[d]$) between populations using a range of summary statistic simulations. The uncertainty in estimates of genetic relative risk across populations, as characterised by the posterior standard deviation s.d.($d$), is presented in Fig 1.

First, we fixed the training sample size, heritability, and genetic distance ($F_{ST}$) and varied the levels of polygenicity. We found that the uncertainty in estimates of genetic relative risk across populations increased sharply with the level of polygenicity. When we simulated summary statistics for a scenario where $N = 100,000$, $h_g^2 = 0.50$ and $F_{ST} = 0.10$ ($F_{ST} = 0.10$ is typical differentiation between divergent continental populations, such as European and African populations), the uncertainty s.d.($d$) ranged from 0.04, 0.15, 0.35 to 0.40 when the proportion of causal variants varied from 0.1%, 1%, 10% to 100%, respectively.

Additionally, the level of uncertainty in estimates of genetic relative risk under an infinitesimal genetic architecture ($p_{\mathrm{causal}} = 100\%$) was relatively high, even at very large sample sizes. Simulating summary statistic data with $p_{\mathrm{causal}} = 100\%$, $h_g^2 = 0.50$ and $F_{ST} = 0.10$, we found that the uncertainty s.d.($d$) decreased from 0.37 to 0.24, as the training sample size increased from 200,000 to 1,000,000.

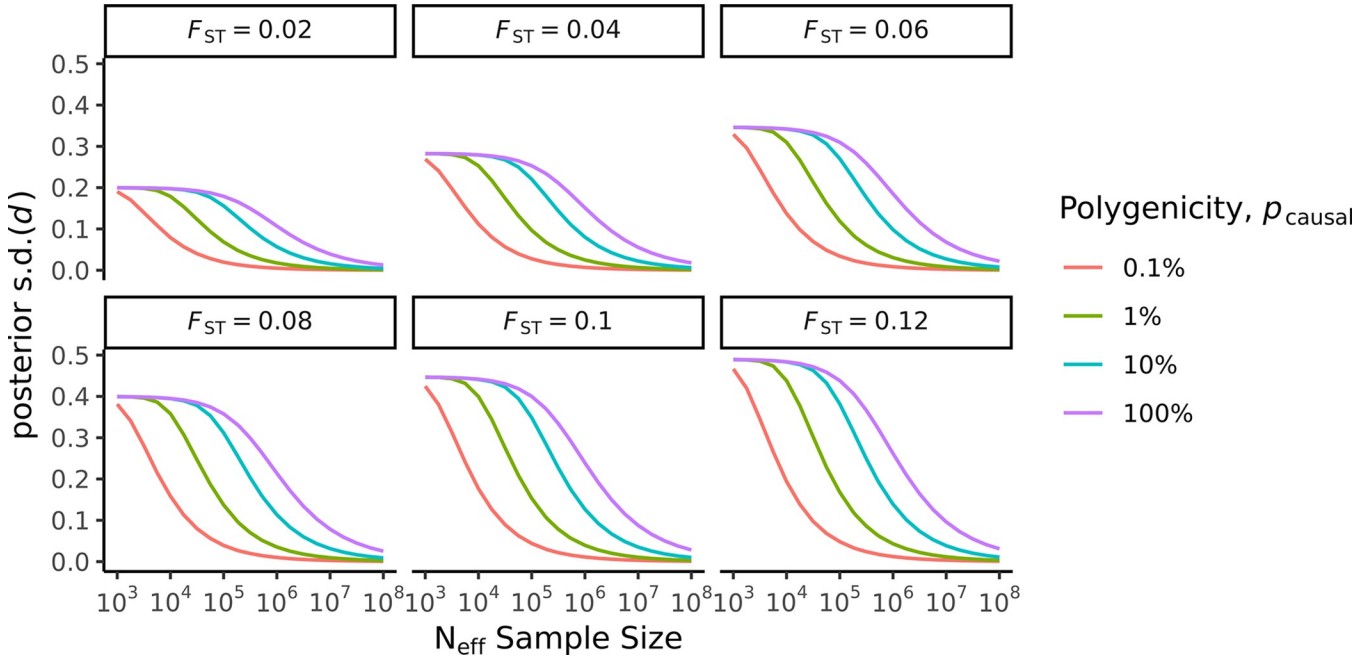

**Fig 1. Estimates of posterior s.d.($d$) based on simulations.** Heritability is held constant at $h_g^2 = 0.50$.

The level of uncertainty s.d.($d$) also increased with genetic distance $F_{ST}$. For example, in the scenario where $N = 100,000$, $h_g^2 = 0.50$ and $p_{causal} = 10\%$, when we varied $F_{ST}$ between 0.02, 0.06 and 0.10, the corresponding uncertainty s.d.($d$) was 0.18, 0.31 and 0.40, respectively.

## Closed-form approximation under infinitesimal model

As shown in Eq (2) above, we further derived a closed-form analytical estimate of the variance of $d = \log(RR)$ under an infinitesimal genetic architecture, where it is assumed that every SNP has effect sizes drawn from $\beta_j \sim N(0, h_g^2/[2Mf_j(1 - f_j)])$ (see methods and S2 and S3 Appendices). We examined the accuracy of this closed-form estimate where the true model was in fact non-infinitesimal, varying both the heritability and polygenicity (Fig 2).

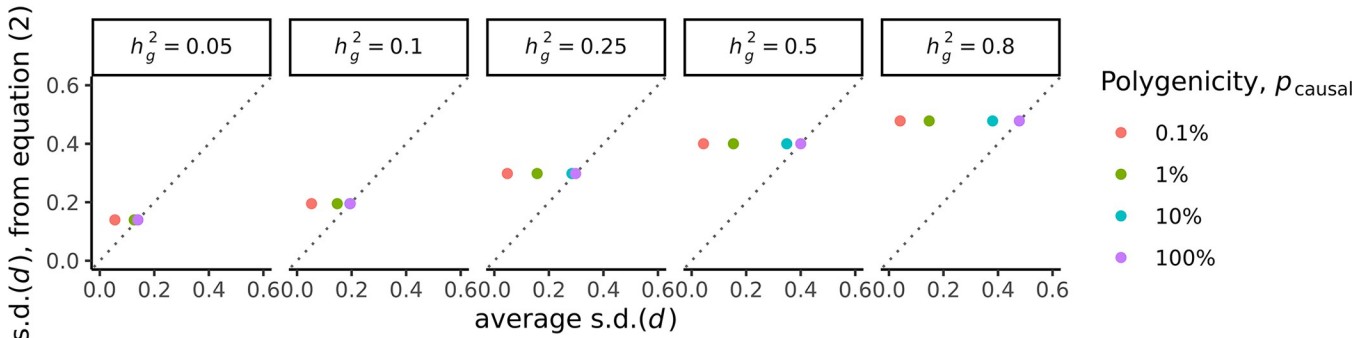

**Fig 2. Closed-form analytical estimator of posterior s.d.($d$) from infinitesimal model compared with average posterior s.d.($d$) from summary statistic simulations.** The x-axis is the average posterior s.d.($d$) from summary statistic simulations. The y-axis is the expected posterior s.d.($d$) computed with Eq (2). Each dot represents the average of 100 simulations replicated for each $p_{causal}$. The number of causal variants was fixed at $M = 200,000$, the genetic differentiation $F_{ST}$ was fixed at 0.10, the GWAS training sample size $N$ was fixed at 100,000.

We found the closed-form estimates were broadly comparable with the expected variance for levels of true polygenicity between 10% and 100%, although there were clear overestimates of the posterior variance when the true polygenicity was below 10%. For instance, for a heritability of $h_g^2 = 0.50$, genetic distance $F_{ST} = 0.10$, and training sample size $N = 100,000$, the closed-form approximation gave s.d.($d$) = 0.40. Meanwhile, when we simulated summary statistics and used the analytic form of the posterior variance per-SNP (see S1 Appendix), we estimated s.d.($d$) as 0.04, 0.15, 0.35 and 0.40 as we varied the polygenicity between 0.1%, 1%, 10% and 100%, respectively.

This suggested that for genetic architectures with low polygenicity, we should instead use analytical estimates of the posterior variance per-SNP (see S1 Appendix) or MCMC simulations to derive the posterior distribution of $d$, rather than the closed-form approximation of Eq (2).

## Assessment of type 1 error, statistical power and bias

Additionally, we assessed the frequentist properties of the Bayesian posterior mean estimator $\hat{d}$, with results presented for a range of scenarios in Figs 3 and 4.

First, having derived a Wald test statistic that accounts for uncertainty relating to the training sample, we demonstrated that this test is well-calibrated, having the expected type 1 error rates (Fig 3). In comparison to our Wald test, we noted that the $t$-test has very high type 1 error rates unless the training sample size is very large. For example, performing a $t$-test using a target sample size of $N = 5,000$ for both of the populations, and assuming a genetic architecture with $h_g^2 = 0.50$, $p_{\text{causal}} = 10\%$ and genetic distance $F_{ST} = 0.10$, we note that a training sample size in excess of $10^8$ is needed to achieve a type 1 error of 0.05. We additionally observed that the $t$-test has the property of being particularly sensitive to scenarios where the target sample size is large relative to the training sample. In these cases, while the training sample may be large enough to provide a precise polygenic score, the $t$-test is still able to detect differences in population means that are due to residual sampling error from the construction of the polygenic score (relating to the training sample size), even when no true difference exists. This leads to an unintuitive finding that the type 1 error can be better controlled for the $t$-test at smaller target samples sizes. Nonetheless, the type 1 error rates for the $t$-test remain uniformly higher than the Wald test (Fig 3).

Second, we estimated the statistical power of the Wald test using simulations, with results for a range of typical scenarios presented in Fig 4A, where we fixed the heritability at $h_g^2 = 0.50$ and the genetic distance at $F_{ST} = 0.10$, and estimated the power for a true relative risk (exp [$d$]) between 1.1, 1.2, 1.5 and 2. We found a strong dependence between the statistical power and proportion of causal variants, $p_{\text{causal}}$. For a training sample size of $N = 100,000$, and assuming a true relative risk of $RR = 1.5$ ($d = \log(1.5)$), the estimated power of the Wald test varied between 1.00, 0.79, 0.18 and 0.07 as the polygenicity increased from 0.1%, 1%, 10% to 100%, respectively. Fixing the polygenicity at $p_{\text{causal}} = 10\%$ (typical of most complex trait genetic architectures), for a training sample size of $N = 100,000$, the power was estimated as 0.11, 0.13, 0.18 and 0.34 as we varied the true relative risk (exp [$d$]) between 1.1, 1.2, 1.5 to 2, respectively. Moreover, for the same polygenicity $p_{\text{causal}} = 10\%$ and a very large training sample size of $N = 1,000,000$, the power was estimated as 0.20, 0.40, 0.91 and 1.00 as we increased the true relative risk (exp [$d$]) between the values 1.1, 1.2, 1.5 and 2, respectively.

Third, we considered the bias and mean square error for the posterior mean estimator $\hat{d}$, showing the bias-variance trade-off for a typical scenario in Fig 4B. For this example, the heritability was fixed at $h_g^2 = 0.50$ and the genetic distance at $F_{ST} = 0.10$, while in this case the true

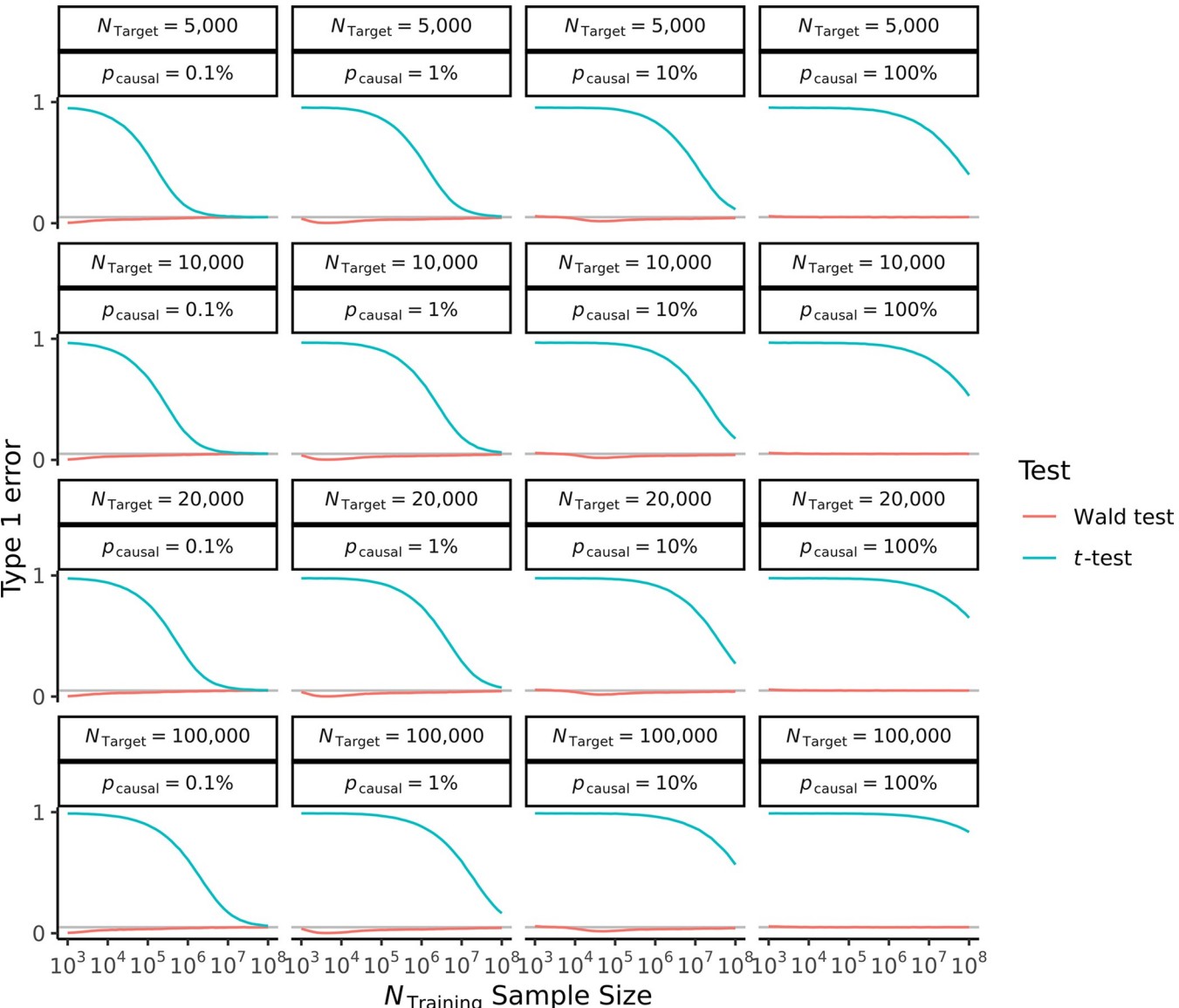

**Fig 3. Type 1 error rates for the Wald test and *t*-test.** Simulation study estimates of the type 1 error rates for a Wald test and *t*-test of size $\alpha = 0.05$. Heritability is held constant at $h_g^2 = 0.50$, genetic distance $F_{ST}$ fixed at 0.10. Polygenicity, and training and target sample sizes are varied as shown. The grey line represents the size $\alpha = 0.05$.

genetic relative risk was fixed at $RR = 1.5$ ($d = \log(1.5)$). We varied the training sample size and the proportion of causal variants $p_{\text{causal}}$. We note that since the posterior mean $\hat{d}$ is a shrinkage estimator, at small training sample size the shrinkage is substantial and the bias very high. Hence, we observed in Fig 4B that the sampling variance of $\hat{d}$ is low at small training sample sizes as the estimates of $\hat{d}$ are drawn towards zero, whilst at the highest training sample size the sampling variance is proportional to $1/N$, and between these extremes at intermediate sample sizes the combination of shrinkage and sampling error results in higher sampling variance. Additionally, the estimator is asymptotically unbiased, with the mean square error consistently converging to zero at large training sample sizes. Moreover, the rate that the bias converges to

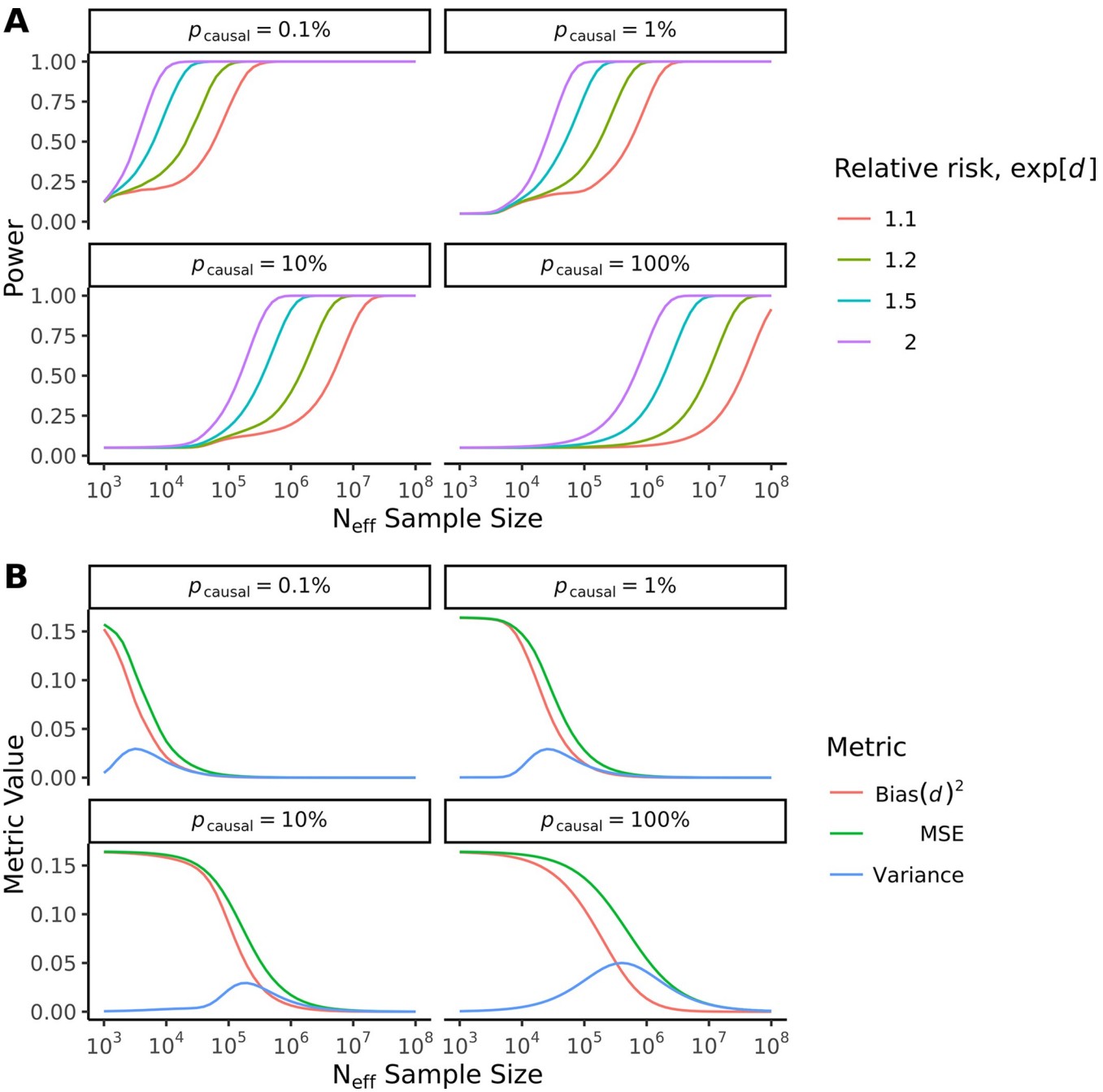

**Fig 4. Frequentist properties of the Bayesian posterior mean $\hat{d}$ for a range of scenarios.** (A) Statistical power for the Wald test with size $\alpha = 0.05$. Heritability is held constant at $h_g^2 = 0.50$, genetic distance $F_{ST}$ fixed at 0.10. (B) Mean square error (MSE), bias (squared), and sampling variance of $\hat{d}$. Heritability is fixed at $h_g^2 = 0.50$, genetic distance $F_{ST}$ at 0.10, and true genetic relative risk $RR = \exp[d]$ fixed at 1.5.

zero is slower for genetic architectures with higher polygenicity $p_{causal}$. For a typical genetic architecture with $p_{causal} = 10\%$, we observe visually that the bias remains high even at training samples of $N = 100,000$, falling rapidly at sample sizes of $N = 1,000,000$ and above.

## Applications

### Case study: Prostate cancer in PRACTICAL Consortium

We provide a real data example based on the analysis performed by Conti et al. [8], which estimated the difference in genetic risk of prostate cancer across European, Hispanic, South Asian, and African populations. We used summary statistics obtained from the multi-ancestry analysis which included 107,247 cases and 127,006 controls. We estimated the mean genetic risks based on allele frequencies from the control groups of each population within the PRACTICAL consortium. We used a set of 1,444,196 HapMap3+ SNPs [17], which we further filtered to a set of 234,822 approximately independent SNPs through clumping using the African (AFR, $N$ = 504) population within the 1000 Genomes project [18] (1KGP), where we clumped based on allele frequency using the bigsnpr R package [19] snp_clumping command with $r^2 <$ 0.1 and 1,000-kb windows.

As we did not have access to a test dataset for parameter tuning, we generated the PGS model using LDpred2-auto [13], also implemented using the bigsnpr R package [19]. This approach estimates hyperparameters from the training data, namely the proportion of causal variants $p_{causal}$, and the SNP-heritability $h_g^2$. We used LDpred2-auto to generate the polygenic score using a Gibbs sampling chain with 3,000 iterations after 1,000 burn-in, with SNP effect sizes averaged across all chains. We used 30 initial values for $p_{causal}$, ranging from $10^{-4}$ to 0.9. We computed the initial $h_g^2$ from the snp_ldsc function. We filtered chains by comparing the scale of the resulting predictions as described in the LDpred2 vignette (bigsnpr, version 1.12.2). Moreover, since the training sample was composed predominantly of European ancestry participants, we used the default EUR reference panel from UK Biobank, provided by Privé et al. [17]. Using the posterior samples $\tilde{\boldsymbol{\beta}}^{(1)}, \tilde{\boldsymbol{\beta}}^{(2)}, \ldots, \tilde{\boldsymbol{\beta}}^{(3,000)}$ (returned using the report_step = 1 argument in the snp_ldpred2_auto command), we generated posterior samples $\tilde{d}^{(1)}, \tilde{d}^{(2)}, \ldots, \tilde{d}^{(3,000)}$ comparing the mean difference in genetic values for European, Hispanic, South Asian and African ancestries, where the allele frequencies were based on the control group in each population. We evaluated the posterior mean and variance of $d$ by averaging across these posterior samples. Finally, using our Wald test, we computed $\chi^2$-statistics and corresponding P-values for comparing the difference in genetic risk across each pair of populations. We also used our simulation approach (S6 Appendix) to evaluate the statistical power available for identifying genetic risk differences of a range of magnitudes (relative risk, $RR$ varied between 1.1, 1.2, 1.5, 2), given the genetic architecture ($h_g^2$ and $p_{causal}$), effective sample size $N$, and genetic distance $F_{ST}$ for each population comparison.

The relative risk estimates across populations are shown in Table 1, where we compared results from our Wald test with those based on the $t$-test analysis performed previously [8]. The hyperparameters were estimated as $h_g^2$ = 0.13 and polygenicity $p_{causal}$ = 0.34%. In comparison to the mean genetic risk for men of European ancestry, we estimated that men of African ancestry had relative risks of 2.99 (95% credible interval, 1.32–6.78, $P$ = 2x10$^{-3}$), while men of East Asian ancestry and Hispanic men had relative risks of 0.46 (95% CI, 0.18–1.19, $P$ = 0.05) and 0.82 (95% CI, 0.60–1.12, $P$ = 0.17), respectively. Hence, we found broadly similar point estimates to the original Conti et al. [8] analysis, with greater uncertainty that reflects accounting for the GWAS training sample size.

We further considered the statistical power available for analysing each of these contrasts using our Wald test, given the observed genetic architecture, effective sample size and genetic distance across populations (Table 1). Using our simulation approach (S6 Appendix), we fixed $h_g^2$ = 0.13, $p_{causal}$ = 0.34, and the effective sample size at $N$ = 232,586. For the comparison of European and African populations, using our derived estimator for $F_{ST}$ (S4 Appendix) which

**Table 1. Comparison of prostate cancer genetic relative risk estimates across populations.**

| Population 1 | Population 2 | Relative genetic risk* | | | | Statistical Power for Wald test | | | | |
|---|---|---|---|---|---|---|---|---|---|---|
| | | t-test, Conti et al. 2021 [8] | | Wald test | | $h_g^2 = 0.13$, $p_{causal} = 0.34\%$, $N = 232,586$ | | | | |
| | | Relative risk (95% Confidence intervals) | P-value** | Relative risk (95% Credible intervals) | P-value | $F_{ST}$*** | $RR = 1.1$ | $RR = 1.2$ | $RR = 1.5$ | $RR = 2.0$ |
| African | European | 2.18 (2.14–2.22) | $<10^{-300}$ | 2.99 (1.32–6.78) | $2 \times 10^{-3}$ | 0.12 | 0.46 | 0.84 | 1.00 | 1.00 |
| East Asian | European | 0.73 (0.71–0.76) | $2 \times 10^{-73}$ | 0.46 (0.18–1.19) | 0.05 | 0.12 | 0.46 | 0.84 | 1.00 | 1.00 |
| Hispanic | European | 0.97 (0.94–1.00) | 0.03 | 0.82 (0.60–1.12) | 0.17 | 0.02 | 0.94 | 1.00 | 1.00 | 1.00 |

Relative risk estimates from Conti et al. 2021 [8] t-test with our Wald test.

* $RR = \exp[d]$, here $d = \mu_1 - \mu_2$, and $\mu_1$ and $\mu_2$ denote the polygenic score mean in population 1 and 2, respectively.

** P-value calculated based on normal approximation on the log-hazard scale using confidence intervals from Conti et al. 2021 [8].

*** $F_{ST}$ was evaluated using our estimator derived in S4 Appendix.

gave $F_{ST} = 0.12$, we estimated the power to be 0.46, 0.84 and 1.00 for identifying a genetic relative risk of 1.1, 1.2 and 1.5, respectively, showing the test to be well-powered. Similar results were found for the comparison of European and East Asian populations, as the population distance was also estimated as $F_{ST} = 0.12$. Additionally, for the comparison of European and Hispanic populations, we found $F_{ST} = 0.02$, which led to estimates of power of 0.94 and 1.00 for identifying a genetic relative risk of 1.1 and 1.2, respectively, hence our Wald test for this contrast was also well-powered.

## Case study: UK Biobank

We provide a further realistic application of our method to assess the impact of the pruning procedure on the accuracy of s.d.($d$). We considered a set of 28 diseases and complex traits from UK Biobank, which we based approximately on sets of phenotypes compiled in previous studies [20,21], which were representative of a diverse range of genetic architectures. We used summary statistics from GWAS conducted from Neale Lab (nealelab.is/uk-biobank), which were derived from $N = 361,194$ individuals of white-British ancestry. For these phenotypes, we evaluated s.d.($d$) using the pruned set of 234,822 SNPs defined above ($r^2 < 0.1$ and 1,000-kb), where we compared the risk differences across European and African populations. We computed s.d.($d$) using LDpred2-auto with 3,000 iterations after 1,000 burn-in. For the hyperparameters $p_{causal}$ and $h_g^2$ estimated using LDpred2-auto, we further computed s.d.($d$) using Eq (1), which assumes SNPs are independent. This enabled us to assess how sensitive our estimates of s.d.($d$) were to any residual correlation that remains across SNPs after the pruning process.

The results for this sensitivity analysis in UK Biobank are presented in Fig 5 and S1 Table. As expected we observed that using Eq (1), which assumes independence between SNPs, there were slightly higher estimates of posterior standard deviation s.d.($d$) than those obtained from the LDpred2-auto approach that accounts for correlation between the SNPs. In particular, using Eq (1), the estimates of s.d.($d$) across 28 phenotypes increased on average by 13.5% in comparison to LDpred2-auto. This shows that our analytical formula leads to slight overestimates of s.d.($d$). Given this finding, our earlier simulation study results (see Verification and Comparison) based on sets of independent rather than pruned SNPs will provide slight overestimates of the true s.d.($d$), and are hence mildly conservative regarding the statistical power and type 1 error rates of the Wald test that we would expect in realistic settings.

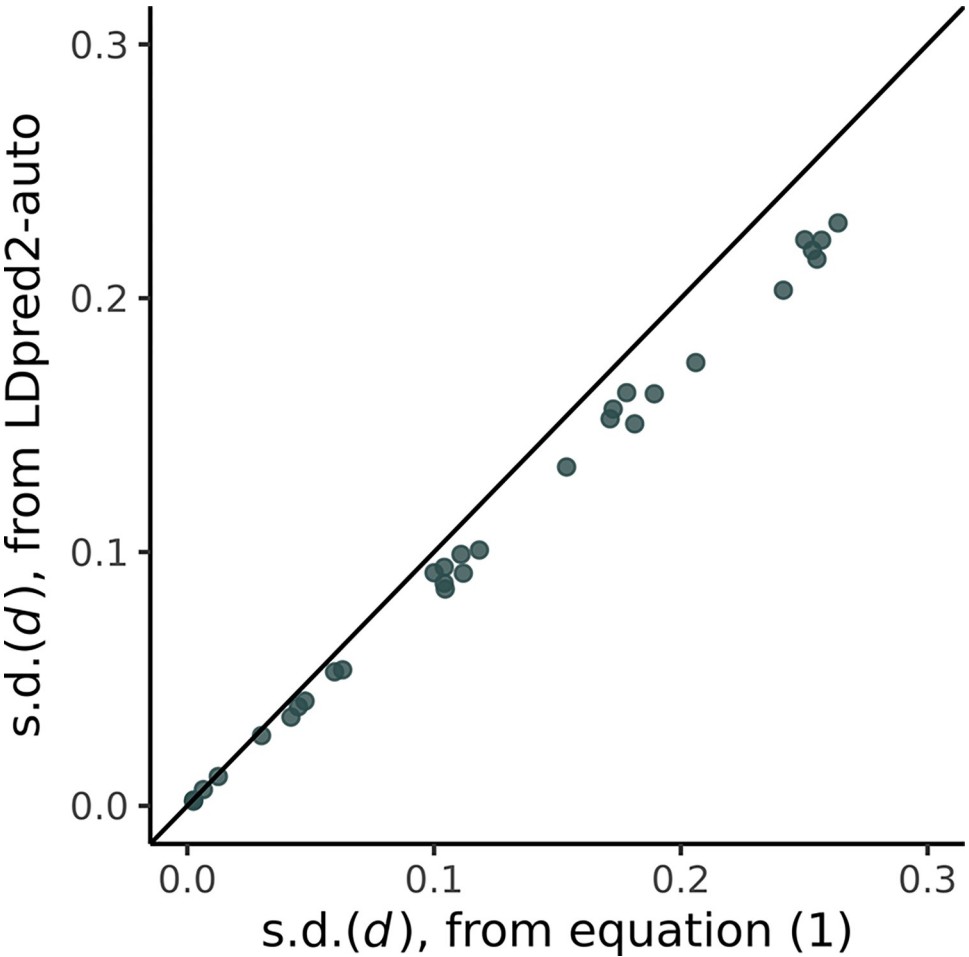

**Fig 5. Validation of posterior variance formula in UK Biobank.** Estimates of the posterior standard deviation s.d.($d$) comparing European and African population disease risk for 28 diseases and complex traits from UK Biobank. For the x-axis, s.d.($d$) is computed under Eq (1), which assumes the SNPs are independent, and for the y-axis, s.d.($d$) is computed using LDpred2-auto which accounts for the correlation between the pruned SNPs.

## Discussion

In this study we develop a framework for understanding the genetic contribution to differences in disease risk across populations. Unlike previous studies, we demonstrate the need to account for uncertainty relating to the size of the GWAS training sample. We further show that uncertainty in estimates of genetic risk differences across populations depends heavily upon the genetic architecture of the phenotype, which is comprised of both the heritability and proportion of causal variants, as well as being dependent upon the genetic distance ($F_{ST}$) between the two populations considered. Moreover, we used a Bayesian framework for estimating relative genetic risk across populations, as Bayesian approaches deliver the highest accuracy for genetic risk prediction [22,23], and are highly flexible in incorporating information about the genetic architecture [24,25].

Using a range of simulation studies, we found a strong relationship between the polygenicity of the genetic architectures and the uncertainty in estimates of genetic relative risk across populations. For phenotypes with polygenicity in the typical range for most diseases and complex traits (between 1% and 10% of variants causal), the uncertainty was low only at sample

sizes >200,000. Hence for most diseases and complex traits, we argue that the sample size present in current biobank-scale GWAS studies and large meta-analyses is too small to clearly discriminate small genetic risk differences between global populations (i.e., relative risk between 1 and 1.5).

For the case of the infinitesimal model, where all variants are assumed to have a causal effect on the phenotype, we further derive a closed-form expression that demonstrates clearly how uncertainty in estimates of relative genetic risk is a function of the heritability, training data sample size, and genetic distance ($F_{ST}$) between populations. We showed that for non-infinitesimal models with a high proportion of causal variants, this simple closed-form expression provided a reasonable estimate of the posterior variance. However, we still recommend the use of either analytical results for the posterior distribution, which we derived, or MCMC simulations to estimate posterior variances, as these approaches provide greater accuracy and are straightforward to implement [13].

We further derived a Wald test statistic relating to the posterior mean estimator of the log relative risk across populations. We demonstrated that this test statistic had well-calibrated type 1 error rates based on simulations, while the power of the test depended strongly upon the proportion of causal variants along with the training sample size. For more polygenic complex trait genetic architectures ($p_{causal} \sim 10\%$), we observed that there was limited statistical power available at current GWAS sample sizes (~100,000) to detect moderate differences in genetic relative risk across populations (relative risk of 1.5 to 2), while to detect smaller differences in risk (relative risk of 1.1 to 1.2) the power only became sufficiently large for sample size more than 1,000,000. In contrast, for less polygenic traits and diseases ($p_{causal} \sim 1\%$), there was in general sufficient power available at current available sample sizes to detect more modest differences in genetic risk across populations, as we also showed in our prostate cancer example. We note further that the conventional $t$-test, which compares mean genetic risk differences by evaluating polygenic scores in separate target samples for each population, has very poorly controlled type 1 error rates, even when the initial training sample size is very large. This is due to the $t$-test detecting mean differences in scores that are due to sampling error in the construction of the polygenic score from the training sample, even when there are no underlying true differences in risk across the populations. This effect becomes more severe as the target sample size increases. This is clearly a problematic property of the $t$-test, and this work provides strong evidence against using this approach.

Having developed a framework for understanding the uncertainty in estimates of relative genetic risk for a general non-infinitesimal model, we provide a real data example by re-analysing the study by Conti et al. [8] on prostate cancer risk. The original analysis by the authors involved computing polygenic scores for prostate cancer risk across samples of participants from several populations, and then using $t$-tests to compare the relative risk across pairs of populations. Such an approach assumes the polygenic scores, which were composed of only 269 variants, capture the complete genetic risk of prostate cancer for individuals without measurement error, which is not a plausible assumption. However, our approach continued to identify a significantly higher genetic risk for males of African compared to European ancestry, and a lower risk for males of East Asian ancestry. Our findings were broadly comparable with the relative risks derived in the original study, with much greater uncertainty surrounding the point estimates, as expected, since uncertainty in the SNP effect estimates arising from the finite GWAS training sample was fully accounted for.

Our study has limitations to note. First, there are several challenges that exist in interpreting differences in polygenic score means across populations. Observed differences could be attributable to various forms of bias, such as uncorrected population stratification [26–28], as well as the choice of discovery data and polygenic score method [29]. While we were unable to

consider the sensitivity of our test statistic to these forms of bias, our best-case scenario provides an informative lower bound on the levels of uncertainty in estimates of population differences in genetic risk.

Second, our approach depends upon using an independent set of variants to model genetic risk differences, avoiding the complexities of modelling linkage disequilibrium patterns across populations. This however can exacerbate the issue of imperfect tagging of causal variants, as information is lost during the pruning process, dampening the predictive performance of the polygenic score. The magnitude of this effect is difficult to ascertain, although we note that LDpred2 estimates have been shown to be robust to the impact of imperfect tagging [12], while only modest reductions in predictive accuracy are lost when pruning markers for linkage disequilibrium [30]. Differences in genetic risk identified by our approach may simply reflect differences in LD with unknown causal variants. We did however try to mitigate the effect of LD differences in the prostate cancer data by using an African (rather than European) ancestry reference panel for the pruning procedure to maximise coverage of causal variants. Nonetheless, we must base our inferences on a set of largely tagged SNPs, and we must further assume that any such allele frequency differences on the tagged SNPs will reflect the frequency differences at the true underlying causal SNPs. Future work should aim to perform simulations and case studies with realistic LD patterns that assess how estimates of population genetic risk differences taken from a set of imperfectly tagged markers, such as the effect estimate on a set of pruned SNPs, compares with the true population difference based on the complete set of causal SNPs.

Third, we have assumed that the causal effect sizes are equal across populations, and hence we focus on differences in disease risk that are attributable to allele frequency differences (i.e., caused by random drift or selection). This is consistent with findings from recent studies which have shown that, when environments are well controlled, there is minimal heterogeneity in causal effects by ancestry [31,32]. However, our approach lacks the flexibility to account for diseases or complex traits that have population-specific genetic architectures, since differences in causal variants [33–35] and effect sizes [36,37] may also potentially drive differences in genetic risk. In these cases, incorporating parameters such as a cross-population genetic correlation [38,39] could be one potential approach, although this would require summary statistics with large sample sizes for at least two populations, which are not always available. Note however that, in our model, it is the product of genotype and effect size that contributes to genetic risk. Thus, if causal effect sizes are truly different, the genetic risk is the same as it would be from variants with the same effect sizes but different allele frequencies. The result of different effect sizes, and also of imperfect tagging, is to increase the effective $F_{ST}$ in our model, and we can thus accommodate these factors to some extent.

Additionally, we have focussed on variability in the training sample, whereas previous studies have only focussed on variability in the target sample. The latter approach is appropriate for comparing a given polygenic score between populations, such as for risk prediction. But for inference about the underlying genetic model it is necessary to allow also for training sample variance. A fuller account would allow for both training and target sample variance. We have not pursued this here as the extension of the Bayesian estimation to a target sample is not trivial; however an ad hoc Wald test could be constructed by summing the sampling variances of the training and target samples. In our prostate cancer analysis the results were essentially unchanged as the variance from the training sample was much greater than from the target sample.

Finally, while our approach can be used to detect true differences due to drift and selection, the impact of other potential sources of population differences, such as non-additive effects [40] and gene-environment interactions [41], and confounding by genetic and environmental effects that correlate with ancestry [42], is unclear.

In summary, we have developed a framework for understanding the genetic contribution to differences in disease risk across populations. We show that uncertainty in estimates of genetic relative risk across populations is highly dependent on the genetic architecture of the disease, particularly the polygenicity, as well as the training GWAS sample size. Accounting for the training sample size, we find evidence that population differences in prostate cancer risk are at least partly attributable to genetic factors.

## Supporting information

**S1 Appendix. Analytical form of posterior mean and variance of SNP effect sizes under non-infinitesimal model.**
(DOCX)

**S2 Appendix. Closed-form expression for posterior variance under infinitesimal model.**
(DOCX)

**S3 Appendix. Wald test on the posterior mean estimator $\hat{d} = \mathbb{E}[d|\hat{\boldsymbol{\beta}}_{\mathrm{GWAS}}; h_g^2, p_{\mathrm{causal}}]$.**
(DOCX)

**S4 Appendix. Estimation of $F_{ST}$.**
(DOCX)

**S5 Appendix. Simulation study steps for estimating the expected $\mathrm{var}[d|\hat{\boldsymbol{\beta}}_{\mathrm{GWAS};}, h_g^2, p_{\mathrm{causal}}]$.**
(DOCX)

**S6 Appendix. Simulation study steps for estimating the bias, sampling variance, and mean square error of $\hat{d}$, and the type 1 and 2 error for the Wald test and $t$-test on $\hat{d}$.**
(DOCX)

**S1 Table. Results from case study in UK Biobank of posterior variance formula.**
(DOCX)

**S1 Fig. Validation of posterior variance formula for $d$ using LDpred2-auto MCMC simulations.** Posterior standard deviation from analytical expressions (Methods and S1 Appendix) compared with estimates from LDpred2-auto using 1,000 MCMC samples after 100 burn-in iterations. $F_{ST}$ was fixed at 0.10. Training sample size was varied between $10^4$ and $10^8$. For each set of parameters, the expected posterior s.d.($d$) was estimated by averaging across 100 simulations.
(TIFF)

**S2 Fig. Validation of closed-form expression for posterior variance under infinitesimal model.** Posterior standard deviation from closed-form expressions for infinitesimal model (S2 Appendix) compared with estimates from LDpred2-grid (with $p_{\mathrm{causal}} = 100\%$) using 1,000 MCMC samples after 100 burn-in iterations. In panels A, B, C and D, each of the parameters $N$, $h_g^2$, $M$ and $F_{ST}$, respectively, are varied while the others are held fixed, with a few scenarios shown in each case. The closed-form expression is plotted as the curved line:

$$\mathrm{var}[d] = \frac{4MF_{ST}}{N}\left(1 + \frac{M}{Nh_g^2}\right)^{-1}.$$
(TIF)

**S1 Spreadsheet. Quantification of data shown in Fig 1.**
(XLSX)

**S2 Spreadsheet. Quantification of data shown in Fig 2.**
(XLSX)

**S3 Spreadsheet. Quantification of data shown in Fig 3.**
(XLSX)

**S4 Spreadsheet. Quantification of data shown in Fig 4.**
(XLSX)

**S5 Spreadsheet. Quantification of data shown in Fig 5.**
(XLSX)

**S6 Spreadsheet. Quantification of data shown in S1 Fig.**
(XLSX)

**S7 Spreadsheet. Quantification of data shown in S2 Fig.**
(XLSX)

**S1 Acknowledgements. Members of the PRACTICAL consortium.**
(DOCX)

## Author Contributions

**Conceptualization:** Iain R. Timmins, Frank Dudbridge.

**Data curation:** Iain R. Timmins.

**Formal analysis:** Iain R. Timmins.

**Funding acquisition:** Frank Dudbridge.

**Investigation:** Frank Dudbridge.

**Methodology:** Iain R. Timmins, Frank Dudbridge.

**Software:** Iain R. Timmins.

**Supervision:** Frank Dudbridge.

**Writing – original draft:** Iain R. Timmins.

**Writing – review & editing:** Iain R. Timmins, Frank Dudbridge.

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
