## [Decision Letter · Decision Letter 0]

1 Mar 2023

Dear Dr Timmins,

Thank you very much for submitting your Research Article entitled 'Bayesian approach to assessing population differences in genetic risk of disease with application to prostate cancer' to PLOS Genetics.

The manuscript was fully evaluated at the editorial level and by independent peer reviewers. The reviewers appreciated the attention to an important problem, but raised many substantial concerns about the current manuscript. Based on the reviews, we will not be able to accept this version of the manuscript, but we would be willing to review a much-revised version. We cannot, of course, promise publication at that time.

If you decide to revise the manuscript for further consideration at PLOS Genetics, please aim to resubmit within the next 60 days, unless it will take extra time to address the concerns of the reviewers, in which case we would appreciate an expected resubmission date by email to plosgenetics@plos.org.

We are sorry that we cannot be more positive about your manuscript at this stage. Please do not hesitate to contact us if you have any concerns or questions.

Yours sincerely,

Michael P. Epstein

Academic Editor

PLOS Genetics

David Balding

Section Editor

PLOS Genetics

Reviewer's **Comments to the Authors:**

Reviewer #1: Timmins presents an interesting method to compare relative genetic risk across populations while accounting for the uncertainty in PGS estimates. He estimates the genetic risk difference and its uncertainty with the MCMC sampling in LDpred2, which is built on a prior work of Ding et al. The author also developed an analytical form of the uncertainty is genetic risk difference estimate under infinitesimal model. It’s also interesting to see that the Fst between two populations contribute to the significance of genetic risk difference. While I appreciate the concept, major concerns should be addressed prior to publication.

Major comments:

1. If the author aims to provide a novel tool for testing the significance of genetic risk difference across population, Type I and Type II error should be assessed.

2. Line 85, is independence assumption required here? I think it’s only useful for deriving the analytical form, since LDpred2 can handle SNPs with LD.

3. For the real data analysis of prostate cancer, more detailed comparison are needed, e.g. provide a table comparing the genetic risk difference estimates from your study and Conti study

4. For the example of prostate cancer, it will be interesting to see the relationship between relative risk and Fst. Does the relative risk (or log relative risk) increases linearly with Fat?

5. There are substantial terminology and mathematical annotations in the main text and supplementary method:

1. Line 81 and 94, what does \\tide{beta} mean?

2. Line 100, 103 and 203 d, instead of \\hat{d} because you are describing the variance of d conditional on the data and hyperparameter

3. Supplementary Line 8, E(\\beta_gwas|\\beta) is mean, not posterior mean, correct term should be E(\\beta|\\\\beta_gwas)

4. Supplementary Line 10, same error for posterior variance

5. Supplementary Line 18, how this \\tide beta differ from \\hat beta_gwas, if they are the same, please use the same annotation

Minor comments:

1. Figure 2, why the analytical form is constant for different polygenicity?

2. Figure 3, I’m a little confused about the curve, is that the posterior distribution of genetic risk difference? If it’s the case, since the EUR is the base population, shouldn’t the relative risk be deterministic 1?

3. Line 174, I’m confused, is there case/control information in 1000 genome?

Reviewer #2: The authors use a Bayesian framework to detect differences in genetic effects between populations. While I agree that the authors use a better approach than what is currently performed in practice, I believe that both approaches are not valid. Indeed, shifts in distributions of PGS across ancestries have been shown in multiple papers, and are likely merely due to random drift of allele frequencies (AFs) of tagging variants used in the PGS (i.e. even if the AFs of a causal are the same, the tagging variant used instead may have different AFs across pops). These small differences accumulate and leads to the shifts in distribution, EVEN WHEN THERE IS NO GENETIC DIFFERENCE BETWEEN THE POPS. It would be hard to convince me otherwise. Another alternative would be to rewrite the paper to clearly show the problem I am mentioning, and warn other people about it.

Other comments:

- L122: the equation is missing the LD

- L153: is there any filtering of the chains before averaging?

- L167-170: you can get the sampling betas directly from LDpred2-auto -> no need for a two-step approach

- I am very surprised there are only 36K variants in common

Reviewer #3: This paper focuses on the problem of identifying the mean difference in the genetic risk scores (PRS) between populations, where the mean difference is denoted as d\\hat in the paper. The paper develops a quantitative approach to calculate the variance of d\\hat, denoted as var(d\\hat). The unique feature of the proposed approach is that it accounts for the uncertainty in the SNP effect size estimates obtained from PRS models. In particular, the proposed approach is based on the assumption that the causal SNP effect sizes are the same between populations. Consequently, the mean difference in PRS between populations can be explained directly by the causal SNP allele frequency differences between populations. As a result, var(d\\hat) becomes a function of the causal SNP allele frequency differences and has a closed form analytic solution under the infinitesimal model where all SNPs are causal. The paper carries out some relatively simple simulations to show that the formula appears to work well.

The overall goal of this paper to account for the uncertainty in SNP effect size estimates for PRS applications is important. However, the paper failed to demonstrate the importance or practical relevance of accounting for such uncertainty, as no comparisons were carried out between the proposed approach and the standard approach where such uncertainty is not accounted for. In addition, the simulation results do not support the closed-form analytic formula derived in the paper, even under the completely polygenic architecture based on which the analytic form is derived. The paper is also not clearly written, and I have a hard time understand the detailed simulation steps. My main concerns are listed below:

1. The main conclusion from the manuscript is that it is important to account for the uncertainty in the SNP effect size estimates when comparing PRS between populations. However, no simulations nor real data results were provided to show that the proposed approach that accounts for such uncertainty works better than the previous approach that directly compares two PRS vectors between populations using a t-test. It is unclear at the moment if one approach provides calibrated type I error control while the other does not, or if one approach is more powerful in detecting PRS differences between population than the other approach. A comprehensive comparison between these approaches should be carried out in both simulations and real datasets to support the conclusion of the paper.

2. One main issue of the paper is that the closed form analytic solution, shown in the last equation on page 6 (line 103), is not supported by the simulations. The close form solution is derived based on the infinitesimal model with 100% polygenicity. In the main simulations (line 210-215), the close-form solution gives an estimate of 0.4 for var(d\\hat). However, the simulations results show that var(d\\hat) is 0.48 under 100% polygenicity. Only when the polygenicity reduces to 10%, var(d\\hat) reduces to 0.42, which is close to the closed-form solution. The manuscript did not explain or explore what sources may have contributed to such deviation. This is an important gap that should be fully investigated.

3. Related to the above point, the closed-form solution is a function of sample size (N), causal SNP number (M), F_st, and heritability h_g^2. Do the simulations support the relationship between var(d\\hat) and each of these parameters as described in the closed-form solution?

4. The main modeling assumption made throughout the manuscript is that the causal effects are the same between populations. With this assumption, the difference in the mean PRS between populations is only due to the differences in causal SNP allele frequencies. However, it is unclear at the moment how realistic this assumption is, given that many previous studies on multi-ancestry modeling make the exact opposite assumption that the causal effect sizes between populations are different. Therefore, it is important to explore how realistic this assumption is in real datasets. It will be important to explore through simulations what consequences are if the causal effect sizes do differ between populations. Ideally, it would be nice if there is a way to quantify the relative contribution of the causal effect size difference and allele frequency difference to the observed PRS difference between populations.

5. I do not follow what was done exactly in the simulations. Specifically, when you compute var(d\\hat), did you use the 2nd equation on page 6 (line 94) for computation? If so, did you first fit a PRS model (e.g. LDpred2) to obtain the estimated \\beta, or did you directly plug in the true beta? If you fitted a PRS model, did you set the hyper-parameters h_g^2 and p_causal to the truth or did you estimate them from the data? Because these two hyper-parameters are estimated in the real data, it would be important to examine through simulations whether the uncertainty in the estimates of these two hyper-parameters influence estimation results on var(d\\hat).

6. In the simulations, the simulated SNPs are independent of each other in both populations. In the real data, however, SNP independence is achieved through pruning. It would be important to explore realistic simulations with realistic LD patterns, where you can further examine how effective the pruning procedure is in obtaining accurate var(d\\hat) estimates.

7. There are standard ways to evaluate the accuracy of variance estimates. Specifically, in the simulations, you can obtain d\\hat in each replicate and then use multiple replicates (e.g. 100 or more) to compute the variance of d\\hat across replicates. On the other hand, you can directly compute var(d\\hat) in each replicate based on your derived formula. You can thus directly compare these two different ways of obtaining var(d\\hat) to examine if the derived formula is accurate or not. This standard approach of evaluating the accuracy of variance estimates should be used in simulations.

**Have all data underlying the figures and results presented in the manuscript been provided?**

Reviewer #1: Yes

Reviewer #2: None

Reviewer #3: None

PLOS authors have the option to publish the peer review history of their article (what does this mean?). If published, this will include your full peer review and any attached files.

Reviewer #1: No

Reviewer #2: No

Reviewer #3: No

---

## [Decision Letter · Decision Letter 1]

13 Sep 2023

Dear Dr Timmins,

Thank you very much for submitting your Research Article entitled 'Bayesian approach to assessing population differences in genetic risk of disease with application to prostate cancer' to PLOS Genetics.

The manuscript was fully evaluated at the editorial level and by independent peer reviewers. While the problem addressed by the manuscript is important, reviewers generally felt that the revision only partially addressed the major concerns raised in the initial submission. Most prominent, two reviewers remain concerned that unobserved causal variants that are tagged by SNPs could negatively impact the method based on the framework’s underlying assumptions. We agree with Reviewer 3 that the impact of the pruning procedure used to address the tagging issue needs to be explored more comprehensively using simulations that incorporate LD. We would be willing to review a further revised version (although we cannot, of course, promise publication at that time) that performs such simulations as well as suitably addressing the other issues raised by the reviewers. Regarding Reviewer 3’s comment about the assumption that causal effects are the same between populations, we note that the recent findings of Hou et al. (Nature Genetics 55: 549-558) might be useful to incorporate into a response and revision.  

If you decide to revise the manuscript for further consideration at PLOS Genetics, please aim to resubmit within the next 60 days, unless it will take extra time to address the concerns of the reviewers, in which case we would appreciate an expected resubmission date by email to plosgenetics@plos.org.

We are sorry that we cannot be more positive about your manuscript at this stage. Please do not hesitate to contact us if you have any concerns or questions.

Yours sincerely,

Michael P. Epstein

Academic Editor

PLOS Genetics

David Balding

Section Editor

PLOS Genetics

Reviewer's **Comments to the Authors:**

Reviewer #1: The authors have addressed most of my concerns. I have some further comments based the author's reply:

1. could you explain the trend of variance in figure 3b, why it goes up and down?

2. page 18, please clarify that you applied your weights to PRATICAL population, also, if possible, please compute the RR estimated directly from the cohort using the case control data, and compare it with the RR estimated from allele frequency difference.

3. could you further comment on how to interpret your results in terms of potential bias of prs as shown in "Testing for differences in polygenic scores in the presence of confounding" https://doi.org/10.1101/2023.03.12.532301

4. the structure of the paper looks confusing, for example, it has two results section, please clearly separate results from methods

Reviewer #2: - "considering a scenario where all SNPs are independent and all causal SNPs are perfectly recorded, and hence no tagging of causal variants is involved" -> This is not realistic at all; unfortunately, I think this will lead to many false positives. There is more and more evidence that genetic effects are similar across populations.

- "We tried to mitigate this possibility in the prostate cancer data by pruning variants using an African ancestry reference panel" -> On the contrary, I think pruning variants only exacerbates the issue because it makes relying on tagging variants even more.

Reviewer #3: While the authors have addressed some of my previous comments, several of my previous key concerns were not well addressed. These key concerns are directly related to the practical usefulness of the proposed method. I re-list these key concerns below, with some additional explanations:

1. The main conclusion from the manuscript is that it is important to account for the uncertainty in the SNP effect size estimates when comparing PRS between populations. However, no simulations nor real data results were provided to show that the proposed approach that accounts for such uncertainty works better than the previous approach that directly compares two PRS vectors between populations using a t-test. It is unclear at the moment if one approach provides calibrated type I error control while the other does not, or if one approach is more powerful in detecting PRS differences between population than the other approach. A comprehensive comparison between these approaches should be carried out in both simulations and real datasets to support the conclusion of the paper.

It will be important to directly compare the previous t-test with the proposed Wald test on the inference of the underlying genetic risk. Specifically, you can examine the type I error control of the t-test in settings where there is no difference in the underlying genetic risk, and examine the power of the t-test if there is a difference in the underlying genetic risk. You can then compare type I error and power of the t-test with the proposed Wald test.

4. The main modeling assumption made throughout the manuscript is that the causal effects are the same between populations. With this assumption, the difference in the mean PRS between populations is only due to the differences in causal SNP allele frequencies. However, it is unclear at the moment how realistic this assumption is, given that many previous studies on multi-ancestry modeling make the exact opposite assumption that the causal effect sizes between populations are different. Therefore, it is important to explore how realistic this assumption is in real datasets. It will be important to explore through simulations what consequences are if the causal effect sizes do differ between populations. Ideally, it would be nice if there is a way to quantify the relative contribution of the causal effect size difference and allele frequency difference to the observed PRS difference between populations.

It is key to assess how good this assumption fits the real-world data, as it determines how useful the proposed method is. Assessing this assumption through both simulations and real datasets is needed and is not beyond the current scope of the paper.

6. In the simulations, the simulated SNPs are independent of each other in both populations. In the real data, however, SNP independence is achieved through pruning. It would be important to explore realistic simulations with realistic LD patterns, where you can further examine how effective the pruning procedure is in obtaining accurate var(d\\hat) estimates.

This is another key issue to address, as it also determines how useful the proposed method is. Assessing this assumption through both simulations and real datasets is needed and is not beyond the current scope of the paper.

**Have all data underlying the figures and results presented in the manuscript been provided?**

Reviewer #1: Yes

Reviewer #2: None

Reviewer #3: Yes

PLOS authors have the option to publish the peer review history of their article (what does this mean?). If published, this will include your full peer review and any attached files.

Reviewer #1: No

Reviewer #2: No

Reviewer #3: No

---

## [Editor Report · Decision Letter 2]

27 Feb 2024

Dear Dr Timmins,

Thank you very much for submitting your Research Article entitled 'Bayesian approach to assessing population differences in genetic risk of disease with application to prostate cancer' to PLOS Genetics. After looking over the revision and your response to previous reviewer comments, we are willing to accept your manuscript provided you add some additional language to the Abstract and Introduction mentioning that your method is only calibrated for dealing with independent SNPs. This will ensure awareness of this issue and perhaps motivate future research in this area. 

1) Upload a Striking Image with a corresponding caption to accompany your manuscript if one is available (either a new image or an existing one from within your manuscript). If this image is judged to be suitable, it may be featured on our website. Images should ideally be high resolution, eye-catching, single panel square images. For examples, please browse our archive. If your image is from someone other than yourself, please ensure that the artist has read and agreed to the terms and conditions of the Creative Commons Attribution License. Note: we cannot publish copyrighted images.

 You can use the link below to log into the system when you are ready to submit a revised version, having first consulted our Submission Checklist.

Yours sincerely,

Michael P. Epstein

Section Editor

PLOS Genetics

---

## [Editor Report · Decision Letter 3]

7 Mar 2024

Dear Dr Timmins,

We are pleased to inform you that your manuscript entitled "Bayesian approach to assessing population differences in genetic risk of disease with application to prostate cancer" has been editorially accepted for publication in PLOS Genetics. Congratulations!

Yours sincerely,

Michael P. Epstein

Section Editor

PLOS Genetics

David Balding

Academic Editor

PLOS Genetics

Comments from the reviewers (if applicable):

**Data Deposition**

http://datadryad.org/submit?journalID=pgenetics&manu=PGENETICS-D-22-01417R3

**Press Queries**

---

## [Editor Report · Acceptance letter]

2 Apr 2024

PGENETICS-D-22-01417R3 

Bayesian approach to assessing population differences in genetic risk of disease with application to prostate cancer 

Dear Dr Timmins, 

We are pleased to inform you that your manuscript entitled "Bayesian approach to assessing population differences in genetic risk of disease with application to prostate cancer" has been formally accepted for publication in PLOS Genetics! Your manuscript is now with our production department and you will be notified of the publication date in due course.

With kind regards,

Anita Estes

PLOS Genetics

On behalf of:
